# Perceptions of non-Western immigrant women on having breast cancer and their experiences with treatment-related changes in body weight and lifestyle: A qualitative study

Anja JThCM de Kruif[1,2]*, Rabab Chrifou[1,3], Ghislaine L. Langeslag[1‡], Annemijn E. C. Sondaal[1‡], Margret M. M. Franssen[1‡], Ellen Kampman[4‡], Renate M. Winkels[4‡], Michiel R. de Boer[1], Marjolein Visser[1], Marjan J. Westerman[1,2]

1 Department of Health Sciences, Faculty of Science, Amsterdam Public Health Research Institute, Vrije Universiteit Amsterdam, Amsterdam, The Netherlands, 2 Department of Epidemiology and Biostatistics, Amsterdam UMC location VUmc, Amsterdam, The Netherlands, 3 Department of Public and Occupational Health, Amsterdam UMC location VUmc, Amsterdam, The Netherlands, 4 Division of Human Nutrition, Wageningen University, Wageningen, The Netherlands

☯ These authors contributed equally to this work.
‡ These authors also contributed equally to this work.
* j.dekruif@amsterdammumc.nl

**Data Availability Statement:** This study analyzes qualitative data and the participants did not consent

## Abstract

### Background

The number of non-Western immigrants with breast cancer in the Netherlands has increased over the past decades and is expected to triple by 2030. Due to insufficient representation in clinical studies, it is unclear what the specific experiences and needs of these women are. Understanding how culture and religion affect these women's experience of breast cancer and how they deal with chemotherapy and treatment-related changes in body weight and lifestyle is crucial for health care professionals to be able to provide effective support.

### Methods

A qualitative study was conducted using semi-structured interviews with 28 immigrant women with a history of breast cancer treated with chemotherapy.

### Results

Women often associated breast cancer with taboo, death or bad luck. Religion offered these women guidance, strength and meaning to the disease, but also limited the women to openly talk about their disease. Women perceived lifestyle factors to have little influence on the development and treatment of cancer. After treatment, however, their thinking changed and these lifestyle factors became of paramount importance to them. They realised that they missed out on information about managing their own diet, exercise and body weight

to have their full transcripts made publicly available. Supporting excerpts from the raw data (quotes from interviews with participants) are available within the text of the paper. Due to ethical restrictions protecting participant confidentiality, the full transcripts of interviews will not be publicly available. Because data contain potentially identifying or sensitive patient information. Instead anonymized excerpts of the transcripts will be available upon request. The data is archived at the VU via the Research Data Management system on Darkstor, facility for sensitive data. A request for access to the data can be made via the corresponding author (Anja de Kruif), via Department of Epidemiology and Biostatistics, Amsterdam UMC location VUmc by j. dekruif@amsterdamumc.nl or via Research Data Management VU via RDM.beta@vu.nl.

**Funding:** This study was funded by the Dutch Cancer Society, htpps//www.KWF.nl (grant numbers UW2011 – 4987 EK and UW2011 -5268 EK) The authors are responsible for the study design, data collection and analysis, discussion, decision to publish, and preparation of the manuscript. The funders had no role in study design, data collection and analysis, decision to publish, or preparation of the manuscript.

**Competing interests:** The authors have declared that no competing interests exist.

and were eager to share their experiences with other women in their culture with newly diagnosed breast cancer.

## Conclusion

Women became aware during and after breast cancer treatment that it was difficult for them to actively deal with their illness under the influence of their culture and religion. Based on their own experiences and acquired knowledge, they would like to give advice to newly diagnosed women on how to deal with breast cancer within their own culture and religion. Their recommendations could be used by mosques, churches, support groups and health care professionals, to ensure interventions during breast cancer treatment meet their religious and cultural needs and thus improve their quality of life.

## Introduction

People with an immigrant background in general have a lower risk of getting cancer compared to the native population. This is due to differences in exposure to risk factors, both in their country of origin and in the new country of residence [1, 2]. This risk can shift after decades, often in the direction of the risk as found in the native-born population [1]. In the last decades, the composition of the population in the Netherlands has changed, partly due to immigration. To illustrate, in 2015, 10% of the population in the Netherlands was foreign-born [3]. Immigrants primarily originate from non-Western countries such as Turkey, Morocco and the former Dutch colonies Indonesia, Surinam and the Netherlands Antilles [3]. It is expected that—in line with the convergence hypothesis [1, 2, 4, 5, 6, 7, 8, 9]—the incidence and mortality rate of cancer among non-Western immigrants will slowly increase and converge with the rates of the native Dutch population. Where breast cancer is concerned, it is expected that by 2030 the proportion of non-Western immigrant women, as part of the total incidence of women with breast cancer in the Netherlands, will have tripled from 2.9% to 8.7% [2].

From most women in Western countries it is known that they perceive breast cancer diagnosis and treatment as a stressful event that threatens all aspects of their lives [10]. It is generally associated with a combination of physical and psychological threats, such as disease-related symptoms and treatment impacts, bodily changes and disruption to employment and social life [11, 12]. Despite the marginal representation of non-Western immigrant women in breast cancer studies, there are indications that differences in perception on breast cancer and breast cancer treatment exist. Studies from across the world highlight that racial and ethnic disparities in breast cancer treatment prevail, for instance amongst African-American and Hispanic women in the USA [13, 14], and women in New Zealand with Maori and Pacific origin [15]. These disparities may contribute to differences in perception and poorer outcomes in women from minorities with breast cancer, probably due to racial discrimination, cultural insensitivity, and poor provider-patient relationships [13, 14, 15].

Diagnosis and treatment of breast cancer can affect body weight, through changes in diet and physical activity. Studies have illustrated that non-Western immigrant women and Western women show differences in dietary intake which may be due to religious, cultural traditions and social barriers relating to food attitudes and body image ideals. Physical limitations due to an acceptance of larger body weights, which is more common in non-Western populations, may influence physical activity negatively [16, 17, 18, 19, 20, 21, 22, 23]. These

differences may lead to higher weight gains during therapy, which is reported to be associated with greater morbidity and poorer survival among breast cancer patients [24, 25]. To illustrate, in studies on the toxicity of chemotherapy for breast cancer, significantly greater weight gain after diagnosis and treatment [26, 27] was found in African American women compared with Caucasian women. African American women felt frustration about weight gain and perceived it as a stressor [28], not being able to control their weight gain because of pain, fatigue, lack of time and knowledge about healthy eating [29], and health concerns [28, 29, 30]. However, they also expressed greater tolerance for larger body sizes [26, 31, 32] and a sense of acceptation of weight gain as a strategy for coping with breast cancer diagnosis [24, 33]. Results from the Women's Healthy Eating and Living (WHEL) study showed that African American breast cancer survivors are more likely than Caucasian survivors to consume more fat, sweets and junk food, and fewer servings of fruits (−0.7/day) [34]. Despite a dietary intervention based on counselling and building on self-efficacy, African-American breast cancer survivors are less successful at making and maintaining dietary changes such as reducing fat and increasing fibre [23] and are less successful at engaging in physical activity [22]. However, in a study among Turkish women, it was shown that weight gain after adjuvant systemic therapy was in line with European and American counterparts [35].

Although breast cancer diagnosis and treatment may have different implications for ethnic groups and are not always culturally tailored, it is not clear to what extent this applies to foreign-born immigrants and the second generation born in the Netherlands. To provide effective support, it is crucial to understand how an immigrant background shapes women´s experiences of breast cancer, and how this may influence coping with chemotherapy treatment and its side-effects, in particular focusing on body weight and lifestyle changes. In most studies women from non-Western ethnic backgrounds appeared to have been under-represented. To reach meaningful and generalizable results a better representation of ethnic minorities in clinical trials is required, because treatment effects might differ according to ethnicity.

In this qualitative study, we explored how non-Western immigrant women experienced being diagnosed with breast cancer, and how they experienced chemotherapy and treatment-related changes in body weight and lifestyle. Through this qualitative approach, we describe how women experienced and dealt with the cultural and contextual factors of cancer diagnosis and treatment in order to contribute to the development of culturally tailored interventions during and after treatment for breast cancer.

## Methods

### Design

A qualitative explorative study was conducted using semi-structured interviews with 28 non-Western first and second-generation immigrant women with breast cancer, treated with chemotherapy. We defined non-Western women as women who originally came from Turkey, Morocco, Surinam and the Netherlands Antilles, the Middle East and Africa. We made an exception for one woman who originally came from Portugal. She took part in our research because of her knowledge gained through her voluntary work for an organization for non-Western women with breast cancer and because of her conversion from Christianity to Islam.

The Medical Ethics Committee of the Wageningen University approved the entire COBRA study (ABR NL40666.081.12).

### Procedure of study sampling

We recruited non-Western immigrant women in the Netherlands who had been diagnosed with non-advanced (I-IIIA) breast cancer and had received either adjuvant or neoadjuvant

chemotherapy. Respondents were recruited via hospitals and general practices in Amsterdam and Rotterdam. It became evident that in practice this would not generate enough respondents. Therefore, organizations that specifically target non-Western immigrant women with breast cancer, and organizations for non-Western immigrant women in general, were approached and agreed to participate in the recruitment of women. Furthermore, through the network of the researchers and snowball sampling, a total of 28 non-Western immigrant women agreed to participate. Four of the respondents from the organisation for non-Western immigrants currently worked as volunteer breast cancer consultants to support women newly diagnosed with breast cancer.

## Data collection

The semi-structured interviews were guided by a topic list [36] based on existing knowledge and complementary literature search. Examples of topics that were included are: the reaction to diagnosis, experiences during and after chemotherapy with weight gain, nutrition and physical health, coping with illness, and the cultural and religious influences on experiences of diagnosis and treatment. We also asked women what advice they would like to give to someone from the same country of origin who is now diagnosed with breast cancer.

The research team reassessed the topic list after the first six interviews. The topic list was expanded with a topic 'the perception on the cause of breast cancer' as the first interviews showed that this perception partly shaped women's reaction after knowing their diagnosis of breast cancer.

Most of the women were interviewed at home by AK, RC, GS and AS; three of the women were interviewed in a quiet room in the hospital where the women had their follow-up appointment. None of the physicians or nurses who treated the patients were present during the interview. During one interview a health care advisor for non-Western patients was present, as this was explicitly requested by the respondent. Furthermore, most of the women were capable to express their feelings and thoughts well in the Dutch language except for two women who were therefore respectively assisted by a friend and a daughter. All women were informed about the study goal prior to the interviews and signed an informed consent. All women were interviewed after their treatment for breast cancer (mean 4.3 y (range 1–12) after diagnosis). The interviews lasted between 30 min and 2 hours (mean 81 min) and were recorded with a voice recorder.

## Data analysis

The verbatim transcribed interviews were analyzed according to a thematic approach [37]. All interview transcripts were summarized and partial first analysis was conducted by AS, GL and MF. Final total analysis was performed through several phases of coding to analyze the data manually after close reading by two researchers (RC, AK). Labels were given to each text fragment of the interview in a system known as open coding. Two interviews were independently coded by two researchers (AK, RC). The findings were discussed via peer-debriefing until consensus was reached. Subsequently, through axial coding the labels were interpreted and clustered into sub-themes and main themes. The next phase of the analysis was selective coding, identifying the essence of each theme, searching for differences and similarities within and across respondents and looking for deviant cases through constant comparison. The framework of themes and the preliminary conclusions based on this analysis were thoroughly discussed among the two researchers (AK, RC), a member of the research team (MW), and with one of the interviewed women (FB). RC as researcher and Muslim woman, and FB as voluntary breast cancer consultant at an organisation for non-Western immigrant women and a

Muslim woman as well, helped to interpret the results and to formulate the needs and recommendations of the women, through their shared knowledge of culture and religion of the respondents. The recommendations describe what had changed for the women after being diagnosed with breast cancer and having undergone treatment, what they would do differently with the knowledge of today and how it will help other women newly diagnosed.

To ensure further validity, the researchers used 'member check', i.e. returning a summary of an interview to a participant to check for accuracy and whether it resonated with their experiences [38], in two ways; 1) ten of the respondents received a summary of the transcript of their interview and were asked if they recognized themselves in what was written and whether they felt their story had been expressed correctly, and 2) with eight other previously interviewed women the results of the interviews were discussed in a group meeting and they were asked if they recognized themselves in the interpretation.

## Results

All twenty-eight women were treated with (neo) adjuvant chemotherapy and surgery and/or radiation and/or hormone therapy. Twenty-three women had migrated to the Netherlands of whom two were refugees (mean 28,1y (range 7–50) and five women were born in the Netherlands. We were able to deduct the religious status of 25 women; 17 identified their religion as Islam, 4 women as other, and 4 women identified themselves as non-religious although three of them were originally Muslim. For further demographic characteristics of the women see Table 1.

The results of the analysis are described in three themes, including needs and recommendations from women for fellow patients, that offer a good illustration of what it means for these women to have breast cancer and to undergo treatment:

1. Role of culture and religion

   - perceptions on cause of breast cancer

   - disclosure of breast cancer diagnosis

   - respondents' needs and recommendations

2. Impact of chemotherapy

   - body weight

   - nutrition

   - physical activity

   - respondents' needs and recommendations

3. Social support

   - partner and family

   - friends

   - fellow patients and consultants

   - respondents' needs and recommendations

In reporting the themes, we used the quotes of the women themselves as much as possible to show their story as extensively as possible.

**Table 1. Demographic characteristics of the study population.**

| Characteristics | | N = 28 |
|---|---|---|
| Age Mean (range) in years | | 45.2 (30–57) |
| Ethnicity N (%) | Turkish | 12 (42%) |
| | Moroccan | 6 (21%) |
| | Surinamese | 5 (18%) |
| | Sudanese | 1 (4%) |
| | Syrian | 1 (4%) |
| | Kurdish | 2 (7%) |
| | Portuguese | 1 (4%) |
| Origin N (%) | Born in the Netherlands | 5 (18%) |
| | Refugees | 2 (7%) |
| | Migrated to the Netherlands | 21 (75%) |
| mean (range) years in the NL | | 28,1 (7–50) |
| Religion N (%) | Islam | 17 (60%) |
| | Hinduism | 3 (11%) |
| | Christianity | 1 (4%) |
| | None | 4 (14%) |
| | Missing | 3 (11%) |
| Marital status N (%) | Married | 22 (78%) |
| | Divorced | 3 (11%) |
| | Engaged | 1 (4%) |
| | Widow | 1 (4%) |
| | Missing | 1 (4%) |
| Children N (%) | Yes | 26 (93%) |
| | No | 2 (7%) |
| Employment status N (%) | Housewife | 7 (25%) |
| | Employed | 15 (53%) |
| | Fired | 5 (18%) |
| | Missing | 1 (4%) |
| | Breast cancer consultant* | 4 |
| Years after diagnosis Mean (range) in years, By 5-year strata N (%) | | 4.2 (1–15) |
| | 1–5 y | 21 (75%) |
| | 6–10 y | 4 (14%) |
| | 11–15 y | 3 (11%) |
| Weight change during chemotherapy Mean (range) in kg, By strata N (%) | | 5.2 (-12 - +28) |
| | -12–0 kg | 4 (14%) |
| | 0–5 kg | 7 (25%) |
| | 6–10 kg | 3 (11%) |
| | >11 kg | 3 (11%) |
| | Unknown | 11 (39%) |
| Weight change after chemotherapy until interview date Mean (range) in kg, By strata N (%) | | 2.1 (-12 - +15) |
| | -12–0 kg | 2 (7%) |
| | 0–5 kg | 10 (35%) |
| | 6–10 kg | 2 (7%) |
| | >11 kg | 3 (11%) |
| | Unknown | 11 (39%) |

(*Continued*)

**Table 1.** (Continued)

| Characteristics | | | N = 28 |
|---|---|---|---|
| Hormone therapy N (%) | | Yes | 15 (53%) |
| | | No | 4 (14%) |
| | | missing | 9 (32%) |

*additional volunteer work

## I. Role of culture and religion

**Perceptions on cause of breast cancer and disclosure of breast cancer diagnosis.** All women experienced breast cancer diagnosis as a very traumatic experience and the word cancer carried such weight that it is not or hardly pronounced. Some women mentioned how breast cancer is viewed in their country of origin; it is the unnamed disease for which there is hardly any treatment. Four women talked about having breast cancer in their country of origin: One of the Surinamese women described cancer as a non-identifiable, bad disease, if you have it you will die. The Sudanese woman mentioned that in her country of origin the word cancer is not pronounced, a doctor will usually not dare to tell the diagnosis to the patient. In Kurdistan, the word cancer is not mentioned because people cannot or do not want to spend money on the treatment of cancer, because the loss of a breast due to treatment for breast cancer is unacceptable. Furthermore, few women can afford the treatment. One of the women from Turkey described that in a less cosmopolitan and more conservative area than Istanbul, cancer is still a taboo. These women were grateful they lived in the Netherlands and could be treated here.

Most of the women explained that their country of origin, their culture, and religion, shaped how they dealt with the diagnosis of breast cancer.

The majority of the women were religious and religion is an important part of their daily life:

*'Faith is a way of life, not only at home, you confess it everywhere. From the moment you wake up you already make a plea to Allah to thank him for waking you up again, and that goes on all day long'* (Moroccan and Muslim woman, 36 y)

Many of the religious women experienced the cause of breast cancer as something that you cannot do anything about, as this is the will of God. This shaped their perspective on the causes of breast cancer:

*'Everything is just for nothing; how healthy you live. I have never smoked, always exercise, always eat healthy, not too much stress. . .yet breast cancer. If you are unlucky in your life or it is in your destiny you get it, you will not prevent it. If God permits, then you get it, and nobody can prevent it, not even the doctors. And if God gives you power or wants you to recover, he always has his reasons, then you get it'* (Turkish and Muslim woman, 48 y)

*'You can't ask why you get breast cancer, that is as being against God'* (Turkish and Muslim woman, 49 y)

Many women struggled with the intentions of God; breast cancer as a test, a punishment:

*'Maybe I am punished, maybe I did something in my life. . . but now I took care of my father in law and my own father and mother, I did a lot of good things'* (Surinamese and Hindu woman, 56 y)

Despite the fact that women experienced breast cancer as a test or a punishment, they believed faith gave them strength and a second chance to come closer to God:

*'I have not been to Mecca, no Quran reading, I am not afraid to die, but I am not yet ready to die. God gave me the strength to keep me going, I am much more aware of faith, praying more, going to wear a headscarf...'* (Turkish and Muslim woman, 33 y)

A minority of the women perceived breast cancer as a result of a polluted environment, eating processed and unhealthy foods, stress, not giving breastfeeding, not having kids, their genetic makeup and also 'bad luck':

*'We're just not living in a society where everything is healthy and natural. So if you think of it, everyone can get it (breast cancer)'* (Turkish and Muslim woman, 40 y)

*'It is genetically in our family, my mother had it, my sister has it, so half of us will get it. Only Allah knew whether I would get it or not'* (Moroccan and Muslim woman, 35 y)

All women, whether religious or not, immediately thought about dying from breast cancer and not being able to care for their children:

*'Cancer is death'* (Turkish and Muslim woman, 49 y)

*'I am not afraid to die, but my children... That hurts, why must I leave, .... afraid of what will happen, my job is not over yet'* (Syrian woman, 52 y, Christian)

**Respondents' needs and recommendations.** Women described how they began to think differently during and after treatment and how that helped them. One of them formulated it as follows:

*'We do not want to know....[..] ... in my community, it is still a taboo, breast cancer as an ordeal of Allah. ... I was a bad person. Now I can tell you these were wrong thoughts. I think it would help some women if it is more open to discussion. There is need to talk about it...'.* (Moroccan and Muslim woman, 36 y)

## II. Impact of chemotherapy

Women reported during and directly after treatment well-known side effects such as; nausea, vomiting, hair loss, loss of energy and fatigue, taste and smell alterations, psychological distress and chemotherapy-related-hospitalizations. Due to these side effects and feelings about diagnosis breast cancer and treatment, women became more aware of the need to take better care of themselves, of their health and their body:

*'If your day has come that is it, you cannot do anything about it, Allah states that. But you must of course take care of yourself and your body. You may not dig your grave and wait on the edge'* (Turkish woman, 41 y)

*'I think I have neglected my body ... I was only busy with the wellbeing of others, for my children, for my family, but you have to start with yourself'* (Turkish woman, 42 y)

Several women struggled with their sexuality and body image during and after treatment:

*'When you get breast cancer your breasts are no longer seen as part of your sexuality, but rather as something that's sick'* (Surinam woman, 54 y)

*'I would turn the light off before I went to the bathroom, did not want to see myself in the mirror: Skin yellow, gray, no hair, no eyebrows. . .. My sister saw me accidentally in the shower and my head was without a hat and she fainted'* (Turkish woman, 49 y)

*'Hair loss is loss of femininity, I am really a man now'* (Moroccan woman, 45 y)

*'I had no breast. . . a large breast right and left breast was empty. That I always had to cover with a scarf. I walked curved, a little hidden yes, . . .. sense of shame, really feeling of shame. . ..'* (Kurdish woman, 48 y)

**Body weight.** Before diagnosis, women thought very differently about the importance of maintaining a healthy body weight. Some women did not worry about their weight:

*'I was not really thinking about it. I am always like this. . . my weight is always up and down. Because I can't keep my hands off yes from the snacking and sweets and I have no regularity in my day. . .'* (Turkish woman, 41 y, BMI >25)

*'Perhaps because I come from Suriname and I know what hunger . . .. uuuh if you are hungry, and it is not there and you come from a family of 10 or 12, then you have to ensure that you get it. That remains your whole life.'* (Surinamese woman, 57 y, BMI 30,8)

While others found it important to watch their weight:

*'I slimmed last year after my third pregnancy to get back to my weight. Yes, my mother thinks that the disease is a result of slimming. . . I said no mom, everybody is slimming. . . No, she said, you should not slim you know. . . I get every time a dressing down from my mother'.* (Turkish woman, 40 y, BMI 23,6)

During treatment many women gained weight because they followed the doctor's advice to eat more:

*'The oncologist had said to me. . . at the first time, we were talking about food, go for good food, eat a lot because you need it. You have to get stronger, because it is just heavy to cope with the next chemo, or it will be postponed. So, uh, well I've taken his advice. I went for the burgers and the chocolate. . .'* (Moroccan woman, 36 y, gained 10 kg weight)

And most of the women were offered help in cooking, sometimes by their husband but mostly by family members as part of their culture to take care of each other. They prepared favorite foods for them:

*'It was difficult for me to eat. . . but my family brought all that food, all my favorites. . .. So, I felt obliged to eat all that food together with them. . ..'* (Syrian woman, 52 y)

Some other women experienced enormous weight loss during chemotherapy:

*'At first, I lost 14 kg during chemotherapy, could barely eat, just drank a lot of water. I received tube feeding from the fourth cycle'.* (Surinamese woman, 57 y)

and also, after chemotherapy: as the same woman indicated:

'Now I started with normal food but my problems are only getting bigger. . .. I am voracious. I have already gained 12 kg in two weeks. A few minutes after a meal I am hungry again. Sometimes I stand in front of the fridge to see what else can I eat, it is so bad. So, I have to pay attention but I don't know how to stop. . .'

After chemotherapy most of the women still struggled with their weight, some until up to three years after chemotherapy as these Turkish women indicated:

*'I really feel like I'm bloated, like a balloon. I have to replace my whole wardrobe. You will never be the same as before your illness. I still cannot wear everything because it is too tight'* (Turkish woman, 49 y, gained 15 kg after chemotherapy)

*'And I do not dare to go on a very strict diet because that was not recommended to me by the doctor'* (Turkish woman, 41 y, gained 10 kg after chemotherapy)

*'I did not mind losing weight, because I do not like having too many pounds. So yes, I actually liked that. But now I do not like it so much that I have gained weight'* (Turkish woman, 35 y, lost 4 kg during chemotherapy, gained 6 kg during hormone therapy)

Only two women called in the help of a dietician, one because of extreme weight loss and the other because of extreme weight gain. They both followed their diet strictly:

*'I was happy, I received a dietary scheme and I ate everything that was on that scheme'* (Sudanese woman, 35 y)

**Nutrition.**   Before diagnosis most women were used to eating a variety of foods, including fruits and vegetables. There were also unhealthy food products in their diets, such as sugary drinks, a lot of red meat and confectionery:

*'I ate a lot of unhealthy food, I realize that now'* (Turkish woman, 53 y)

This Syrian woman found it important to watch her diet already before diagnosis:

*'. . .and I eat healthy, but little. I don't eat three times a day, but I eat a good sandwich with cheese. I also drink milk, I always eat healthy vegetables and fruit. I don't smoke, I don't drink, and I eat no fat'* (Syrian woman, 52 y)

During chemotherapy the majority of the women mentioned their diet changed due to chemotherapy and lacked variety because of their symptoms:

*'Only spinach with bread. Every day, hot spinach with pepper, my sister-in-law makes the best with homemade bread. That is the only thing I eat. If I ate other things, it brings up again to the mouth, but spinach remains in my belly'* (Moroccan woman, 45 y, gained 20 kg weight during chemotherapy)

Despite bad appetite due to smell and taste problems, some women felt hungry, but the only food they really liked was fatty food:

*'I had such a desire in a noodle snack and asked my friend to bring me five, I ate three, and a croquette, and the doctors were very pleased that I had eaten'.* (Turkish woman, 30 y)

Muslim women experienced the Ramadan as an important aspect of their faith. They struggled with their desire to participate in the Ramadan as usual, but they realized children, older adults and the sick are exempt from the Ramadan because of their health. Despite that, they sometimes did have a feeling of guilt:

*'If you are sick, you have to think about your health, think about your body. But now I walk with guilt. . . because I have always participated in the Ramadan until now, from my twelfth I always did'* (Turkish woman, 48 y)

Doctors and nurses reacted differently on the women's desire to participate in the Ramadan:

*'I told the doctors that I must, I must adhere to Ramadan and fast, I was used to it from the age of ten, as a little girl. That doctor told me, don't do it, you just have to eat and you need your medication, it's better for you. I said doctor, listen, if I don't, then I feel that I betray my faith'* (Moroccan woman, 45 y)

*'I wanted to try Ramadan and if it did not work, I would stop. I could actually get meds and everything through the drip the doctor gave me. And I felt really good, fasting isn't harmful, it is good for your body, it is purifying for your body'* (Moroccan woman, 36 y)

After treatment a small majority of the women mentioned that they changed their diet, because they became more conscious of their eating habits and tried to decrease the amount of unhealthy foods in their diets:

*'I try not to cook with deep-frozen products or ready-to-serve dinners but I try to make everything fresh by myself. I try. . . yes, it is not always easy . . . to eat vegetables and drink organic milk. I am more aware of healthy food now, snacks are really poison for me'.* (Moroccan woman, 36 y)

'I now drink my tea without sugar. I try not to drink soft drinks. I am afraid that the cancer comes back. They say that sugar and long-life food is not good. . ..'. (Turkish woman, 33 y)

But sometimes they experienced problems eating healthy foods during frequent family visits:

*". . .. all those extra delicacies they made specially for me . . ..'* (Turkish woman, 48 y)

**Physical activity.**   Before diagnosis women had different tasks during the day. They sometimes combined household chores, taking care of the children (especially if they were young) with a job. Some of them had also taken on the task of caring for their (sick) parents. About half of the women felt that their days were completely filled with household chores and work:

*'My daily activity consisted of my household chores and working as a cleaning lady, that was exercise for three hours a day, going back and forth, this was enough physical activity for me'* (Turkish woman, 56 y)

A few women mentioned that their lives take place in and around the house. They usually stay at home and only go outside for a reason:

*'I only go outside if I have an appointment or if I want to go shopping'* (Sudanese woman,

35 y)

However, the other half of the women described how they were doing sports next to household chores and raising their children:

*'I went three times a week to the gym. I worked six hours a day. I started at nine and was home at three o'clock, I was constantly busy; I can't remember that I was at home on an afternoon'* (Turkish woman, 48 y)

During treatment the majority of the women were too tired to be physically active and they were sometimes not even able to go outside due to the tiredness they felt.

*'. . . so quickly exhausted. . . I went to visit my father at the end of the street. At one point they brought me by car, I had no strength left, I was just tired and I was literally gasping'* (Moroccan woman, 36 y)

*'I was very weak during chemotherapy, I was just lying on the sofa, I had no energy, no feeling'* (Syrian woman, 52 y)

For some women, the chemotherapy did not have much impact on their physical activity. They felt weak, for a couple of days at the most, but after that, they would continue their normal daily activities.

'*The first day of chemo, I was brought home, the second day I couldn't cycle, but I was able to continue with my normal daily activities the third day.*' (Portuguese woman, 44y)

*'. . ..at the end of the chemo, my husband drove beside me in the car and I rode the bike. He thought it was really ridiculous that I was still on the bike. I just have to go. I thought it's just not healthy if you get a whole bag of poison through your body and you just sit or lie down. You should move, you just have to stay active, it just has to be, it just has to flow through your body'* (Turkish woman, 54 y)

However, they could not perform intensive physical activities such as working every day or going to the gym three times a week.

Directly after treatment, hardly any of the women were able to be as active as they were before the treatment. Women still suffered from tiredness, fuzziness and they sometimes felt lazy. Some women picked up their normal daily activity routine, but did so in a less intensive manner:

*'After the chemo, I started to go to the gym again, but just two times a week and for half an hour'* (Surinamese woman, 40 y)

For some women, the tiredness gradually diminished. However, for other women, fatigue continues to affect their daily lives and they feel incapable of physical activity:

*'10 years after chemotherapy, I still get tired very quickly. I get tired quickly after physical strain'* (Kurdish woman, 48 y)

Just a few of the women were able to pick up their normal daily activities as previously.
Several women indicated they were confronted with too many emotional events such as the death of a son, spouse, mother or divorce, which meant that physical activity was totally disregarded.

**Respondents' needs and recommendations.**   Women indicated that changes in diet and exercise during and after chemotherapy require information:

*'After chemotherapy I learned it is important what I eat and that exercising is very important. People have to know! Probably via church or breast cancer group.'* (Syrian woman, 52 y, Christian)

And, about the importance of taking good care of yourself:

*'If your husband is ill, you just have to take care of him, if your child is ill, your mother…. It is a matter of course in our culture. But if you get sick yourself, you always tried to take care of all the rest. It is a matter of course in our culture… What really helps is to know now that you have to put yourself in first place and to take care of yourself, then you can really take care of others'.* (Moroccan and Muslim woman, 36 y)

They indicated the importance of being able to understand and discuss questions and uncertainties:

*'Really, if you want to mean something for migrants and you want to take them seriously, you have to offer them an explanation, this has to do with life and death. And I also notice that the doctors do not tell the whole story to the children who go along as interpreters, because they are afraid to scare these children. And the children do not always want to tell the whole story to their mother. The interpreter is no longer there because of the cutbacks. Those interpreters simply have to return'.* (Turkish and Muslim woman, 53 y)

## III. Social support

**Partner and family.**   Social support was identified as important for all women. In particular, many married women experienced their husbands as providing the greatest support. The emotional support of the immediate and extended family during and after the treatment was also a source of hope and help to endure this period.

Partners responded in many different ways to the emotions and needs of their wives suffering with breast cancer: with hope and support, but also with a feeling of helplessness:

*'My husband gave me hope, told me that it will be okay, that I must be strong and fight* (Turkish woman, 49 y)

*'…. he looked really in your eyes you know…. he tried not to make it stand out, but I noticed it as if he is also suffering with me. Well he was very sorry but he tried to support me so much'* (Turkish woman, 40 y)

*'My husband has had a lot of patience with me, I don't think all Moroccan men are like that. My husband is like that, he does not talk much about it. He leaves it here, put it in his heart.*

*But if he starts talking, then I understand. He says, of course, I want to remove the burden for you, but I cannot'.* (Moroccan woman, 45 y)

A few of these women mentioned that breast cancer made them feel vulnerable in their relationships and told their partners they did not have to stay with them:

*'I would understand if he leaves such a sick woman. Have given him the space but he does not want to, how can I leave you if you are sick? But I cannot guarantee that I will be a beautiful woman again in 3 years'.* (Syrian woman, 52 y)

*'I have a man, but I do not have children, that is not bad. That will come soon, and if it does not come, then no. . . . . .yes then I do not mind. My husband can marry other women and have children, then I will get peace of mind'* (Sudanese women, 35 y)

Some women experienced little to no support because partners did not realize the seriousness of the disease or did not understand that women were trying to deal with it in a positive way. Three women were abandoned by their partner because they could no longer talk about cancer and have sex with each other:

*'My husband apparently took it lightly as if I had a flu, did not understand that I had a lot of grief'* (Turkish woman, 33 y)

*'He said to me: I do not understand why you laugh, I do not understand why you have joy in your life, you have cancer, you are dying, do you realize that?'* (Turkish woman, 30 y)

*'He did not expect me to do no cooking anymore and that the house was not always clean, he never did anything. . . . and suddenly he came home with another woman'* (Surinamese woman, 40y).

*He left me after 28 years of marriage because I became ill and he could not tolerate my illness. If he had cancer or another disease, I would never do this to him. Never. I would've stayed with him with all my love. I would care for him. My heart is broken into a thousand pieces. Even cancer didn't cause me so much pain; this is the most difficult part.'* (Kurdish woman, 48 y)

Women did not want to burden their families, as they already felt helpless. They said they would hide their feelings from their family and would only take care of themselves after they had fulfilled all their duties towards their husbands and children. That is why some of them sometimes wanted to go to their appointments alone and be sick on their own:

*'I just sent my husband to his work and I only wanted to cry when the children were in school or already in bed'* (Turkish woman, 41 y, 3 kids)

Families were sometimes very shocked about the breast cancer diagnosis and often did not know much about cancer. Women could not talk openly with their family about their illness because cancer is a taboo and some women felt ashamed of having cancer:

*'I had terrible reactions from my family, they were busy but mainly concerned with their own fear and sadness. . . I thought I am the cancer patient but it is only about you. . .'* (Turkish woman, 54 y, widow)

*'People did not dare to come because cancer is taboo, even my own family did not want to hear about it'* (Sudanese woman, 35 y)

*'For my family it was new, I was the first in the family to have cancer so they were all very upset and could not talk about it. Much later it turned out that two cousins also had breast cancer, they never told anything about it. They thought that you should be ashamed of having had cancer, and that disappointed me'* (Surinamese woman, 54 y)

**Friends.**   From their community, women experienced support. Paying visits to people who are ill is a social and religious obligation for Muslims. Many of the Islamic respondents talked about receiving a lot of visitors during their treatment:

*'My sister was here, my neighbours were kind to me, my friends. Everyone was really nice to me. Yes, really. . . They also came to bring food. They are such lovely people. The whole room was full of flowers. Some people, so many people had come, that really helps you. You are not alone. I had visit, after visit, after visit, and I was really happy with it..'.* (Turkish and Muslim woman, 48 y)

Some women experienced the community support negatively:

*'I did not need it. I don't want to be called or visited. Only my sisters knew, no one else. No, I do not want to be called every time, I'm not waiting for that and that happens when everyone knows'* (Moroccan and Muslim woman 36 y)

Not all of the women had a reliable social network they could depend on, due to emigration to the Netherlands. Most family members, who could provide support, still lived in their home country. Due to the vast distance and the financial costs, these women were not able to receive support from their relatives:

*'My family is in Turkey, luckily, I have my female friends, every day one of them is with me, and I can discuss everything with them . . . I really had to talk about it'* (Turkish and Muslim woman, 40 y)

In addition, women indicated that female friends had an important role in support:

*'It is easier to share this with female friends than with family'* (Turkish woman, 35 y, no religion)

**Fellow patients and consultants.**   Women had different opinions about contact with their fellow patients. Some women said they enjoyed the company of their 'chemo buddies' and liked exchanging information with them:

*'. . .. talking with other women with breast cancer was very good, it reduced the pain'* (Sudanese woman, 35 y)

Other women explained they did not like talking with fellow patients nor going to a support group, because these sessions put them in a negative mind-set.

Several women said they came into contact with Mammarosa (a contact group for women with breast cancer women for whom Dutch is not their mother tongue) in various ways:

*'I actually did not have people to share until I ended up at Mammarosa. My family cannot understand me because they have not felt it, they know that I am sick but they cannot feel it for me. Fellow patients can feel the same . . .'* (Turkish woman, 53 y)

**Respondents' needs and recommendations.** Social support worked in two different ways. As this Turkish woman said: consciously look for the support you need from your family:

*'You really need that support, you know. And I received a lot from my cousins, nephews, my brothers, my sisters, my parents, my friends. Really everyone was with me, it had stimulated me enormously. Because you think of everything, is it okay with me? Do they tell me everything? And if someone from your family says, oh, it's okay, come on, you can do it. You can handle it. . . you need such people and have to ask for support'.* (Turkish woman, 40y)

Or if talking openly about breast cancer within your family is not possible, find other people who can give you what you need, sometimes that is a contact group for women with breast cancer (Mammarosa):

*'Cancer is a taboo in Sudanese culture. Nobody talks about cancer, cancer means death. I only had my husband, who also did not want to talk about it. During chemo I ended up at Mammarosa, met other women and could talk about it; 'What do you have, and what is the side effect? And how can I continue with that? Talking was very good it reduced my fears and I got more rest because I spoke to other women with the same disease at Mammarosa'.* (Sudanese woman, 35 y)

The woman who had converted from Christianity to Islam, explained how she learned to invite women for information about breast cancer, as one of the volunteer breast cancer consultants at Mammarosa:

*'We recently gave information about breast cancer in the mosque as volunteers from Mammarosa. The meeting was widely advertised and, in the end, it was a handful of women came. When I talked about breast cancer, they looked at me with so much shock in their eyes. I asked what's going on here? And they said: ". . . don't mention the word. . . because if you hear about it and read and know then you bring it upon yourself . . ..". So, if you want to give information about breast cancer you have to place it under the heading of health education for women'.* (Portuguese woman, 44y)

## Discussion

In this qualitative study, we found that experiences of non-Western women with breast cancer during diagnosis and treatment were often influenced by religion and culture. Many women perceived the cause of breast cancer as the will of God, punishment, as a test, or bad luck. However, faith also gave strength and made it possible to come closer to God. For most women cancer is still synonymous with dying and their greatest fear is leaving behind their children and

missing the opportunity to raise them. Besides the well-known side effects of treatment, women also reported weight gain as a frequent and lasting struggle even in the period after chemotherapy. The advice of health care professionals on weight gain, whether or not to participate in Ramadan during treatment, and on physical activity, were sometimes misinterpreted by women due to their communication skills and language barriers. Social support was given by spouses, family, friends and fellow patients. Visits from family and friends, based on the social and religious obligation for Muslims to visit those who are ill, were not always appreciated. This sometimes resulted in family members and friends not being told about the breast cancer diagnosis, especially when breast cancer was indicated as a taboo or something to feel ashamed of. Conversely, other women talked openly about their disease to family members and friends and needed these visits, especially from family members. Contact with fellow patients was not always experienced positively, as the stories of others gave a negative mindset. However, other women felt supported through the support group for breast cancer patients. Women became aware that it was difficult for them to actively deal with their illness due to their culture and religion. They felt they were unable to exert much if any influence on the development and treatment of their cancer. After treatment, however, they realised that they had missed out on information about managing their own diet, exercise and body weight and were eager to share their experiences with other women with newly diagnosed breast cancer in their culture. They would like mosques, churches, support groups and health workers to adopt their recommendations in order to meet the religious and cultural needs of non-western women during breast cancer treatment, thereby improving their quality of life.

The majority of women (17 of 28) in our study are Muslim, and their perceptions of breast cancer and breast cancer treatment were often shaped by their religion and culture. However, for the non-religious women culture also influenced their perceptions of breast cancer and breast cancer treatment. This religious and cultural influence on the perceptions of breast cancer and breast cancer treatment is confirmed by studies exploring the reactions of Turkish women to their cancer diagnosis [39, 40, 41]. These studies found that cancer was perceived as death and suffering [40], family support was very important but inadequate [41], and difficulty to accept illness negatively affected quality of life [41].

Almost half of the women in our study gained weight during and or after treatment. Women experienced problems with dietary intake and physical exercise as part of their weight gain. They viewed weight gain as a source of distress and as a health concern. While this finding is consistent with prior research [28, 29, 30], our results also highlight that some women viewed weight loss as a positive experience. Women were frustrated they were not able to control changes in their weight and experienced these weight changes as unrelated to their eating or activity behaviours. Our results are in line with a recent qualitative study with African-American breast cancer survivors, which demonstrated that women experience difficulty maintaining weight control efforts because of barriers such as pain, fatigue, lack of time, lack of knowledge about what to eat and how to prepare foods in a healthy way [29]. In addition, African-American breast cancer survivors have been shown to be less likely than Caucasian survivors to report that they changed their diet or increased their physical activity following diagnosis and treatment [24]. This may be because of a greater tolerance for larger body sizes among African-American women [31, 32]. This could be the same among the Surinamese and older Turkish and Moroccan women in our study. Perhaps it is different for younger Turkish women born in the Netherlands, as they seem to think it is important to watch their weight before diagnosis but also during treatment.

This study shows the influence of culture and religion on the women's perceptions of chemotherapy and treatment-related changes in body weight and lifestyle. Studies examining the behaviour of women in response to these treatment related changes show that not many

women, including non-Western women, succeed in positive long-term behavioural change [42, 43, 44]. With the growing number of cancer survivors, there is a need for effective behavioural interventions. To our knowledge, this is one of the few studies in which non-Western women reflected on their experiences of breast cancer and breast cancer treatment in order to made recommendations available for effective interventions of cancer patients within minority populations.

## Implications for practice

The recommendations of, in particular, Muslim women are numerous and valuable. These recommendations are based on their own needs and experiences during the treatment of breast cancer. These recommendations can be considered more or less in accordance with the Self-Determination Theory (SDT) [45]. Muslim women indicated that there should be more open talk about breast cancer and not in terms of punishment or penance. They want to make their own conscious decision to live more healthily, they want to be able to make autonomous choices. They need more time and more information from health care professionals about healthy food choices and encouragement to exercise more. That is in line with their own religious values and norms, such as the requirement in the Quran that you must take good care of your own body. More than half of the women contacted others such as fellow sufferers through support groups to help them with their own experiences, while the other women do not want contact with others. Further development of these recommendations in line with SDT provides starting points for interventions to support these women and clearly contributes to a better quality of life during and after chemotherapy. We describe their recommendations as implications for three different groups: 1) all people around them, 2) breast cancer support groups, and 3) health care professionals. These groups will be able to support women with newly diagnosed breast cancer by taking responsibility for these recommendations and helping with the implementation (see Box 1).

### BOX 1. Implications for practice

For all network members:

- *Help women to create opportunities for talking openly about breast cancer*

- *Help women to take good care of themselves, and not only take care of others first*

For breast cancer support groups:

- *Beware that some women have a negative attitude towards breast cancer due to stories they heard from fellow patients*

- *Continue to create a meeting place for women of all kinds of ethnic minorities for support, information, and an opportunity to share stories*

- *Encourage women to eat healthy and exercise and maintain a healthy body weight during and after treatment*

For health care professionals:

- *Create more time during doctor's visits and ensure that interpreters are available so that patients can fully understand the consequences of a breast cancer diagnosis and its treatment*

- *Help women to control their body weight due to barriers such as fatigue, and lack of knowledge. Help women understand what they need to eat, how to prepare food in a healthy way and encourage them to exercise during and after treatment*

- *Understand the religious and cultural obligations Muslims have to visit ill people because that means larger group of visitors during hospital stay. Most women were happy to have their family with them*

Important for all patient groups and health care professionals:

*Information about breast cancer can best be included in a health education package for women; breast cancer does not have to be explicitly mentioned.*

## Strength and limitations

The strength of this study lies in the diversity of the stories told. The interviewed women had different ethnic backgrounds and religions, and they had experienced a variety of breast cancer treatments. These interviews were thus very rich and informative, offering a great variety of insights.

The approach and methods were chosen carefully. A patient-expert, one of the interviewed women converted from Christianity to Islam, and one of the researchers (RC) helped to interpret the results especially in relation to the meaning within Islam, in order to better understand the women and their struggles, and to gain insight into the recommendations, beliefs and perceptions of these women.

The primary limitations of this study are related to the method of sampling. The recruitment of women took a lot of time and effort. That is why convenience sampling was chosen as the method of sampling. This meant that women who were easiest to find, ended up in our sample. Despite this, it was a diverse sample with women who were very open in telling their story. As mentioned above, this is one of the strengths of the study. However, if subsequent studies are meant to convey something about a specific group, such as groups with the same religion or the same cultural background, it is recommended that ample time be allowed for recruitment and that a more heterogeneous sampling method is chosen. The fact that only women with a reasonable command of the Dutch language were selected could mean that the sample represents only a part of the non-Western immigrant population of women with breast cancer, namely women who were already better integrated into Dutch society.

## Conclusions

Women became aware during and after breast cancer treatment that it was difficult for them to actively deal with their illness due to their culture and religion. At diagnosis, most women perceived lifestyle factors to have little influence on the development and treatment of cancer. After treatment, however, their thinking changed and these lifestyle factors became of paramount importance to these women. They realised that they missed out on information about optimal diet, exercise and body weight during and after treatment. Based on their own experiences and acquired knowledge, they are eager to give advice on how to deal with breast cancer to newly diagnosed women within their own culture and religion. A set of recommendations was formulated to support health care professionals, breast cancer support groups, mosques and churches to meet the religious and cultural needs of non-western women during breast cancer treatment and thus improve their quality of life.

## Acknowledgments

We thank all participants for their time and their openness in participating in this research. Also, we would like to thank Rebecca Rendle-Buehring for her help in preparing the manuscript. Special thanks to Lide van der Vegt and Fatima Batista, Mammarosa Den Haag, and Ina Speelman, Mammarosa Amsterdam.

## Author Contributions

**Conceptualization:** Anja JThCM de Kruif.

**Data curation:** Anja JThCM de Kruif.

**Formal analysis:** Anja JThCM de Kruif, Rabab Chrifou, Ghislaine L. Langeslag, Annemijn E. C. Sondaal, Marjan J. Westerman.

**Funding acquisition:** Ellen Kampman, Renate M. Winkels.

**Investigation:** Anja JThCM de Kruif, Rabab Chrifou, Ghislaine L. Langeslag, Annemijn E. C. Sondaal, Margret M. M. Franssen.

**Methodology:** Anja JThCM de Kruif, Rabab Chrifou.

**Project administration:** Anja JThCM de Kruif.

**Resources:** Anja JThCM de Kruif.

**Software:** Anja JThCM de Kruif.

**Supervision:** Ellen Kampman, Renate M. Winkels, Michiel R. de Boer, Marjolein Visser, Marjan J. Westerman.

**Validation:** Anja JThCM de Kruif.

**Visualization:** Anja JThCM de Kruif.

**Writing – original draft:** Anja JThCM de Kruif, Marjan J. Westerman.

**Writing – review & editing:** Anja JThCM de Kruif, Rabab Chrifou, Ghislaine L. Langeslag, Annemijn E. C. Sondaal, Margret M. M. Franssen, Ellen Kampman, Renate M. Winkels, Michiel R. de Boer, Marjolein Visser, Marjan J. Westerman.

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
