## [Decision Letter · Decision Letter 0]

12 Dec 2019

PONE-D-19-21419

Perceptions of non-Western immigrant women on having breast cancer and their experiences with treatment related changes in body weight and lifestyle: a qualitative study

PLOS ONE

Dear Dr de Kruif,

Thank you for submitting your manuscript to PLOS ONE. After careful consideration, we feel that it has merit but does not fully meet PLOS ONE’s publication criteria as it currently stands. Therefore, we invite you to submit a revised version of the manuscript that addresses the points raised during the review process.

Please revise your manuscript in line with reviewer comments.  In particular, please remove Boxes and titles as these are results not recommendations.

We would appreciate receiving your revised manuscript by 12 January 2020. To enhance the reproducibility of your results, we recommend that if applicable you deposit your laboratory protocols in protocols.io, where a protocol can be assigned its own identifier (DOI) such that it can be cited independently in the future. For instructions see: http://journals.plos.org/plosone/s/submission-guidelines#loc-laboratory-protocols

We look forward to receiving your revised manuscript.

Kind regards,

Rosemary Frey

Academic Editor

PLOS ONE

Journal Requirements:

Reviewers' comments:

Reviewer's Responses to Questions

**Comments to the Author**

1. Is the manuscript technically sound, and do the data support the conclusions?

Reviewer #1: Yes

Reviewer #2: Yes

2. Has the statistical analysis been performed appropriately and rigorously? 

Reviewer #1: Yes

Reviewer #2: N/A

3. Have the authors made all data underlying the findings in their manuscript fully available?

Reviewer #1: Yes

Reviewer #2: No

4. Is the manuscript presented in an intelligible fashion and written in standard English?

Reviewer #1: Yes

Reviewer #2: Yes

5. Review Comments to the Author

Reviewer #1: The paper explored the experience of non-Western immigrant women from diagnosis to treatment of breast cancer and described the recommendation from the qualitative interview study. With an increased number of migrants in Western countries, it is timely to explore those minority group’s perception and experiences on cancer to develop a proper intervention to order to improve their quality of life.

For the abstract, can you add subheadings to assist readers?

The introduction showed a fairly good argument; however, some paragraphs lack details and the flow.

For example, the first paragraph started with the increasing immigrant’s number then moved to the risk of cancer. The second and third paragraph are very short, with 2sentences and 3 sentences, respectively. Can you elaborate these paragraphs to build a story? Fourth paragraph does not flow well with the previous paragraph. Rewrite the fourth paragraph as you started with weight gain then suddenly moved to dietary intake. There must be another sentence to link the next sentence.

In page 4, line 92-3, punctuation should be changed from ‘.’ To ‘,’. Otherwise the sentence of ‘in particular’ is an incomplete sentence.

The manuscript has good details of methods and data analysis for a qualitative study.

The result is quite lengthy, but one of thing I don’t understand is what those recommendation boxes for. I originally thought it would be the summary of recommendation, but it is still just transcribing the interview with the highlight. I assume you have presented those recommendations in boxes as they are important findings in your study. However, the current presentation looks no different from other themes.

Table 1 might be better to re-structure to assist readers. I would suggest presenting variables under each characteristic. It would also be good to add % of sample next to the number.

Page 11 line 221, ‘Each description of the three themes closes with a box with these recommendations’ Wouldn’t it be better to change as ‘ each description of the three themes presented in a box with…..’ ?

In your discussion, you have summarised the study well at the beginning, but what is the significance of your study? There have been many studies showing that religion and culture have a large impact on women with breast cancer from diagnosis. Please state the significance of your study result. Some of the paragraphs in the discussion could be reorganised or re-written to have a clear topic sentence at the beginning and the summary sentence at the end. At this stage, your discussion does not tell a good story.

Reviewer #2: Thank you for the opportunity to review this paper, it is very well written and interesting. I have only a few minor comments.

Line 169: it may be helpful to provide a little bit more explanation about the meaning of 'member check'.

Line 577: There's no participant identifier with the quote.

Line 794: I don't think 'representative' is the right term to use in relation to qualitative studies, as generalisability is not the goal of this type of research.

6. PLOS authors have the option to publish the peer review history of their article (what does this mean?). If published, this will include your full peer review and any attached files.

Reviewer #1: No

Reviewer #2: No

---

## [Author Response · Author response to Decision Letter 0]

21 Jan 2020

To: Rosemary Frey

Academic Editor

PLOS ONE

Dear Editor,

We would like to thank you and the reviewers for assessing our paper, entitled “Perceptions of non-Western immigrant women on having breast cancer and their experiences with treatment related changes in body weight and lifestyle: a qualitative study”.

The extensive review, from a variety of viewpoints, has definitely helped to strengthen our paper. We have written a point-by-point response to the comments of the reviewers below, which details the modifications we have made in the revised manuscript. We hope that these revisions will make the manuscript suitable for publication in your journal. 

On behalf of all authors,

Anja de Kruif

Reviewer #1: The paper explored the experience of non-Western immigrant women from diagnosis to treatment of breast cancer and described the recommendation from the qualitative interview study. With an increased number of migrants in Western countries, it is timely to explore those minority group’s perception and experiences on cancer to develop a proper intervention to order to improve their quality of life.

Response: Thank you for your positive comments on our study.

For the abstract, can you add subheadings to assist readers?

Response: We agree that adding subheadings will definitively assist the readers. We have added these to the abstract and have expanded the text within the guidelines of Plos One and incorporated the significance of the study in the abstract, based on the suggestion of the reviewer. We hope this will more clearly show what the study entails.

The introduction showed a fairly good argument; however, some paragraphs lack details and the flow.

For example, the first paragraph started with the increasing immigrant’s number then moved to the risk of cancer. The second and third paragraph are very short, with 2 sentences and 3 sentences, respectively. Can you elaborate these paragraphs to build a story? Fourth paragraph does not flow well with the previous paragraph. Rewrite the fourth paragraph as you started with weight gain then suddenly moved to dietary intake. There must be another sentence to link the next sentence.

In page 4, line 92-3, punctuation should be changed from ‘.’ To ‘,’. Otherwise the sentence of ‘in particular’ is an incomplete sentence.

Response: Thank you very much for your clear remarks. We have rewritten the introduction based on your suggestions.

The manuscript has good details of methods and data analysis for a qualitative study.

Response: Thank you very much for your positive comments.

The result is quite lengthy, but one of thing I don’t understand is what those recommendation boxes for. I originally thought it would be the summary of recommendation, but it is still just transcribing the interview with the highlight. I assume you have presented those recommendations in boxes as they are important findings in your study. However, the current presentation looks no different from other themes.

Response: The boxes were meant as important findings and we apologize if this was not very clear. We therefore removed the boxes and placed these texts under the three themes as the respondents’ needs and recommendations (line 306-313; line 502-522; line 632-660) to better reflect the significance of the study and the structure of the article. In the discussion, our recommendations for the specific target groups based on the respondents’ needs and recommendations have been placed in one box in the ‘implications for practice’ section (line 753-766). 

Table 1 might be better to re-structure to assist readers. I would suggest presenting variables under each characteristic. It would also be good to add % of sample next to the number.

Response: Thank you, we have restructured table 1 (from line 224).

Page 11 line 221, ‘Each description of the three themes closes with a box with these recommendations’ Wouldn’t it be better to change as ‘ each description of the three themes presented in a box with…..’ ?

Response: Based on your previous comment, we have now removed the boxes. 

In your discussion, you have summarised the study well at the beginning, but what is the significance of your study? There have been many studies showing that religion and culture have a large impact on women with breast cancer from diagnosis. Please state the significance of your study result. Some of the paragraphs in the discussion could be reorganised or re-written to have a clear topic sentence at the beginning and the summary sentence at the end. At this stage, your discussion does not tell a good story.

Response: Thank you very much for this remark, it has helped us to better highlight the relevance of the study. We now emphasized the relevance of the study as much as possible in the abstract, in the introduction, in the methods (line 182-186) and in the summary at the beginning of the discussion (Line 680-689) and later on in the discussion (line 717-724). We hope the discussion now tells the story well.

Reviewer #2: Thank you for the opportunity to review this paper, it is very well written and interesting. I have only a few minor comments.

Response: Thank you very much for your positive comments.

Line 169: it may be helpful to provide a little bit more explanation about the meaning of 'member check'.

Response: in line 187-189 we explain the meaning of member check.

Line 577: There's no participant identifier with the quote.

Response: we have added a participant identifier to each quote (Line 565-566). 

Line 794: I don't think 'representative' is the right term to use in relation to qualitative studies, as generalisability is not the goal of this type of research.

Response: We agree and have replaced the word representative for heterogeneous in line 785

---

## [Decision Letter · Decision Letter 1]

22 Jun 2020

Perceptions of non-Western immigrant women on having breast cancer and their experiences with treatment related changes in body weight and lifestyle: a qualitative study

PONE-D-19-21419R1

Dear Dr. de Kruif,

We’re pleased to inform you that your manuscript has been judged scientifically suitable for publication and will be formally accepted for publication once it meets all outstanding technical requirements.

Kind regards,

Rosemary Frey

Academic Editor

PLOS ONE

Additional Editor Comments (optional):

Reviewers' comments:

Reviewer's Responses to Questions

**Comments to the Author**

1. If the authors have adequately addressed your comments raised in a previous round of review and you feel that this manuscript is now acceptable for publication, you may indicate that here to bypass the “Comments to the Author” section, enter your conflict of interest statement in the “Confidential to Editor” section, and submit your "Accept" recommendation.

Reviewer #2: All comments have been addressed

2. Is the manuscript technically sound, and do the data support the conclusions?

Reviewer #2: Yes

3. Has the statistical analysis been performed appropriately and rigorously? 

Reviewer #2: N/A

4. Have the authors made all data underlying the findings in their manuscript fully available?

Reviewer #2: No

5. Is the manuscript presented in an intelligible fashion and written in standard English?

Reviewer #2: Yes

6. Review Comments to the Author

Reviewer #2: I am satisfied that the authors have addressed all reviewer suggestions, and have improved the manuscript where necessary. It's a great paper.

7. PLOS authors have the option to publish the peer review history of their article (what does this mean?). If published, this will include your full peer review and any attached files.

Reviewer #2: No

---

## [Editor Report · Acceptance letter]

25 Jun 2020

PONE-D-19-21419R1 

Perceptions of non-Western immigrant women on having breast cancer and their experiences with treatment-related changes in body weight and lifestyle: a qualitative study 

Dear Dr. de Kruif:

I'm pleased to inform you that your manuscript has been deemed suitable for publication in PLOS ONE. Congratulations! Your manuscript is now with our production department. 

Kind regards, 

on behalf of

Dr. Rosemary Frey 

Academic Editor

PLOS ONE